# Lenticulostriate Vasculopathy in Very-Low-Birth-Weight Preterm Infants: A Longitudinal Cohort Study

**DOI:** 10.3390/children8121166

**Published:** 2021-12-09

**Authors:** Yi-Li Hung, Chung-Min Shen, Kun-Long Hung, Wu-Shiun Hsieh

**Affiliations:** 1Department of Pediatrics, Cathay General Hospital, Taipei 10630, Taiwan; b82401103@yahoo.com.tw (Y.-L.H.); shen8471@yahoo.com.tw (C.-M.S.); 2School of medicine, Fu-Jen Catholic University, New Taipei City 242, Taiwan; klhung@ms10.hinet.net; 3Department of Pediatrics, Fu-Jen Catholic University Hospital, New Taipei City 243, Taiwan; 4Department of Pediatrics, National Taiwan University Children’s Hospital, Taipei 100, Taiwan; 5Department of Pediatrics, National Taiwan University College of Medicine, Taipei 100, Taiwan

**Keywords:** lenticulostriate vasculopathy, prematurity, very-low-birth-weight, cranial ultrasound

## Abstract

Background: The pathogenesis and clinical significance of lenticulostriate vasculopathy (LSV) are unclear. Our study aimed to determine the prevalence, presentation, and evolution of LSV, and the perinatal risk factors associated with LSV among very-low-birth-weight (VLBW) preterm infants. Methods: One-hundred-and-thirty VLBW preterm infants were retrospectively enrolled in this study. Serial cranial ultrasound examinations were performed regularly from birth until a corrected age of 1 year. Infants with LSV were assigned to early-onset (≤10 postnatal days) and late-onset (>10 postnatal days) groups. Data describing the infants’ perinatal characteristics, placental histopathology, and neonatal morbidities were collected, and the groups were compared. Results: Of the VLBW infants, 39.2% had LSV before they were 1 year old. Linear-type LSV was the most common presentation, and >50% of the infants had bilateral involvement. LSV was first detected at 112 ± 83 postnatal days, and its detection timing correlated negatively with gestational age (GA) (R^2^ = 0.153, *p* = 0.005) and persisted for 6 months on average. The infants with and without LSV had similar perinatal characteristics, placental pathologies, cytomegalovirus infection rates, and clinical morbidities. The late-onset LSV group comprised 45 (88.2%) infants who had a significantly higher rate of being small for gestational age (SGA) and used oxygen for longer than the infants without LSV. After adjusting a multivariable regression model for GA and SGA, analysis showed that the duration of oxygen usage was an independent risk factor for late-onset LSV development in VLBW infants (odds ratio: 1.030, *p* = 0.032). Conclusion: LSV may be a nonspecific marker of perinatal insult to the developing brains of preterm infants. Prolonged postnatal oxygen usage may predispose VLBW preterm infants to late-onset LSV development. The long-term clinical impacts of LSV should be clarified.

## 1. Introduction

Lenticulostriate vasculopathy (LSV) refers to hyperechogenic vessels detected in neonates’ thalami and basal ganglia, using cranial ultrasound (cUS). Growing awareness of LSV over the past 10 years has led to demonstrations of its associations with a variety of fetal and neonatal diseases, which can antenatally or postnatally affect the developing brain [1]. Grant et al. described LSV as branching echogenic structures on cUS images from a neonate with cytomegalovirus (CMV) infection [1]. Teele et al. reported on 12 patients with LSV caused by congenital infections or trisomy 13 and described LSV as a deposition of basophilic material in the arterial walls supplying the basal ganglia and thalamus [2].

The incidence of LSV in preterm neonates varies from 2.2% to 32% [3,4]. Mittendorf et al. showed that a high antenatal dose MgSO_4_ was associated with LSV in preterm neonates in a large cohort study [5]. Various morbidities, such as low Apgar score, bronchopulmonary dysplasia (BPD), and bacteremia, have also been reported to be correlated with LSV [3,6]. However, most studies have focused on serial cUS examinations of LSV evolution in preterm neonates [5,7,8]; data from serial epidemiological LSV follow-up studies involving cUS scanning of very-low-birth-weight (VLBW) preterm neonates are lacking and no consensus exists regarding perinatal risk factors associated with LSV development. If LSV indicates a cerebral insult, preterm infants may be more vulnerable to LSV development because more complex perinatal events occur in immature brains. This study aimed to determine the prevalence, presentation, and evolution of LSV, and the perinatal risk factors associated with LSV among VLBW preterm infants before a corrected age of 1 year.

## 2. Materials and Methods

This 7-year retrospective, observational cohort study involved preterm neonates who were admitted to the neonatal intensive care unit (NICU) of Cathay General Hospital in Taipei between January 2011 and March 2018, and whose birth body weights (BBW) were <1500 g. Neonates with major congenital or chromosomal abnormalities, and those who died in the NICU, were excluded. All neonates underwent regular cUS scanning, from birth until a corrected age of 1 year. At our institute, VLBW preterm neonates are routinely scheduled to undergo cUS on postnatal days (PNDs) 1, 3, 7, and 21, and then every 2 weeks until discharge. After discharge, cUS examinations are scheduled every 2 months until a corrected age of 1 year. Well-trained technical staff who were blinded to the enrolled neonates’ perinatal characteristics performed the cUS, using an HD 11 Ultrasound System (Philips Medical Systems, Bothell, WA, USA) and a 6 MHz curved-array transducer. The cUS examinations were conducted through the anterior fontanelle, and standard images of the coronal and sagittal planes were captured, as previously described [9], and were stored in a picture archiving and communication system. A neonatologist initially interpreted the images and then an experienced pediatric neurologist confirmed and reported the findings.

LSV is defined as the presence of punctate, linear, or branching hyperechogenic lesions in the basal ganglia–thalamic region in coronal and parasagittal views. An LSV diagnosis was confirmed by an independent neonatologist and neurologist. The presence of other major cranial lesions, including intraventricular hemorrhage and periventricular leukomalacia, were also recorded. We used Hemachandra’s classification to separate the neonates into early-onset and late-onset LSV groups [10]. Early-onset LSV was defined as LSV documented before or on the tenth day of life, which suggests that the triggering events occurred before delivery, and late-onset LSV was defined as LSV first seen after the tenth day of life; it is associated with perinatal or postnatal risk events.

To determine LSV-associated risk factors, maternal and neonatal clinical characteristics were recorded. Maternal risk factors included preeclampsia, gestational diabetes mellitus (GDM), preterm premature rupture of the membranes (PPROM) >18 h, delivery mode, antenatal steroids and MgSO_4_ usage, C-reactive protein (CRP), and blood white blood cell (WBC) counts before delivery, and the presence of histological chorioamnionitis (HCAM). Two independent pathologists performed pathologic examinations of the placentas, and HCAM severity was graded according to Redline et al. [11]. The neonatal risk factors included gestational age (GA), BBW, sex, small for GA (SGA), and large for GA, which were based on domestic nationwide definitions [12,13], an Apgar score <7 at 5 min, respiratory distress syndrome (RDS) [14] (≥grade III which required ventilator support with surfactant therapy), hemodynamically significant patent ductus arteriosus (PDA) [14], persistent pulmonary hypertension of the newborn (PPHN) treated with nitric oxide (iNO) [15], pneumothorax, early- or late-onset sepsis [16], BPD, which was defined as the need for oxygen at 36 weeks postconception [17], necrotizing enterocolitis (NEC), which was defined as grades 2–3 using Bell’s modified criteria [18], retinopathy of prematurity (ROP) ≥ stage III [19], and the duration of oxygen use or mechanical ventilation. The presence of a CMV infection was defined as serum that was positive for CMV-specific immunoglobulin IgM or a positive urine CMV isolation. The hospital’s ethics committee approved the study (CGH-P108117).

### Statistical Analyses

Relationships between PND, postmenstrual age (PMA), and GA, and the first LSV detection were examined using linear regression analyses. Infants with and without LSV and early- or late-onset LSV were compared regarding demographic and perinatal variables using chi-squared analyses or Fisher’s exact test for categorical variables and an independent sample *t*-test or the Mann–Whitney-U test for continuous variables. To examine the factors associated with LSV development, univariate logistic regression analyses were undertaken and significant variables and the GA were included in the multivariate logistic regression model. The odds ratios (ORs) and 95% confidence intervals (CIs) were calculated to estimate the relative risks of LSV. The statistical analyses were performed using SPSS (version 15.0) for Windows (SPSS Inc., Chicago, IL, USA). A value of *p* < 0.05 was considered significant.

## 3. Results

During the study period, 168 preterm neonates who weighed <1500 g were admitted to the NICU. Three neonates had multiple congenital anomalies and were excluded. Thirty-one neonates died during hospitalization, and four were lost to follow-up. Finally, 130 neonates were enrolled in the study, and their mean GA was 28.2 ± 2.3 weeks and the was 1097 ± 247 g. Seventy-two (55.4%) neonates were boys.

Of the 130 VLBW enrolled neonates, 51 (39.2%) were diagnosed with LSV based on ≥1 cUS examination from birth until a corrected age of 1 year. LSV presented as linear (Figure 1A), branching (Figure 1B), and punctate (Figure 1C) type lesions. Linear LSV was the most common presentation in 56.9% of the neonates, and seven (13.7%) neonates had punctate LSV. Thirty neonates (58.8%) had bilateral LSV at the first detection. Right involvement was more common than left involvement among the neonates with unilateral LSV. LSV was first detected at a mean age of 112 ± 83 PNDs (range, 1–371 PNDs) and at a mean PMA of 44 ± 11.5 weeks. The PND on which LSV was first detected and the GA were negatively correlated (Figure 2). The PMA at which LSV was first detected did not correlate with the GA. Most LSVs remained until almost 6 months after birth, and then disappeared by a mean corrected age of 7.3 ± 3.6 months. LSV disappearance on particular PNDs or at particular PMAs was not associated with GA, or BBW analyzed by linear regression.

Early-onset LSV was first detected in 11.8% (6 of 51) of neonates within 10 days of life, and late-onset LSV was present in 88.2% (45 of 51) of neonates. Table 1 shows the presentations and evolution of early- and late-onset LSV. The LSV appearance period was longer in the late-onset LSV group than in the early-onset LSV group. The age at which LSV disappeared was also older in neonates with late-onset LSV compared with early-onset LSV. The groups did not differ regarding the LSV appearance, LSV laterality, and associated abnormal cUS findings.

Table 2 summarizes the perinatal characteristics of VLBW neonates with and without LSV, and the LSV-associated risk factors, including placental pathology findings and postnatal complications. Infants with LSV had a significantly higher rate (42.9%) of intermediate-to-advanced HCAM than those without LSV (24.6%) (*p* = 0.045). Infants with LSV used oxygen for longer durations than neonates without LSV; however, the multivariate logistic regression analyses that were adjusted for GA and the LSV-associated covariates listed in Table 2 did not determine any independent risk factors associated with LSV occurrence (Table 3). A total of 22 of 51 neonates with LSV underwent CMV evaluations, and four (18.2%) had CMV infections. A total of 14 of 79 neonates without LSV underwent CMV evaluations, and three (21.4%) had CMV infections.

BBW, birth body weight; BPD, bronchopulmonary dysplasia; CMV, cytomegalovirus; CRP, C-reactive protein; GA, gestational age; GDM, gestational diabetes mellitus; HCAM, histological chorioamnionitis; IVH, intraventricular hemorrhage; LGA, large for gestational age; LSV, lenticulostriate vasculopathy; NEC, necrotizing enterocolitis; PDA, patent ductus arteriosus; PPHN, persistent pulmonary hypertension of the newborn; PPROM, preterm premature rupture of the membranes; PVL, periventricular leukomalacia; RDS, respiratory distress syndrome; SGA, small for gestational age; ROP, retinopathy of prematurity; WBC, white blood cell.

Table 4 presents the perinatal characteristics of neonates with early- and late-onset LSV. The groups did not differ regarding the maternal and neonatal characteristics. The preterm infants with late-onset LSV had a significantly higher SGA rate and a significantly longer oxygen usage duration than neonates without LSV. After adjusting for GA and SGA, multivariate logistic regression analyses showed that oxygen usage duration was independently associated with late-onset LSV (aOR, 1.030, 95% CI, 1.003–1.058, *p* = 0.032). (Appendix A).

BBW, birth body weight; BPD, bronchopulmonary dysplasia; CMV, cytomegalovirus; GA, gestational age; GDM, gestational diabetes mellitus; HCAM, histological chorioamnionitis; LGA, large for gestational age; LSV, lenticulostriate vasculopathy; NEC, necrotizing enterocolitis; PDA, patent ductus arteriosus; PPROM, preterm premature rupture of the membranes; RDS, respiratory distress syndrome; SGA, small for gestational age; ROP, retinopathy of prematurity.

## 4. Discussion

This 7-year retrospective, observational cohort study enrolled all VLBW preterm infants born at a single institution. The neonates underwent frequent cUS for longer than infants involved in previous studies of LSV [5,7,8]. Therefore, we recruited all preterm neonates whose LSV showed prenatal or postnatal insults, and we evaluated the incidence and the evolution of LSV in VLBW preterm infants longitudinally for 1 year. In our study, we found that LSV is not an uncommon finding of preterm VLBW and is persistent. Prolonged oxygen usage rather than CMV infection predisposes preterm VLBW infants to LSV development.

The precise incidence of LSV in preterm neonates is unclear, because of poor interrater agreement regarding LSV identification [3]. In our study, the incidence of LSV in the VLBW preterm neonates was 39.2%, which is much higher than previously reported rates [3,5,7,8,10]. The preterm neonates in this study underwent serial cUS examinations until a corrected age of 1 year, which enabled us to recruit all neonates with LSV over a longer period of time, and this may underlie the higher LSV incidence in this study compared with rates reported by previous studies that involved single evaluations. Sisman et al. reported LSV rates of 0.9%, 2.2%, and 9.6% on PNDs 1–4, 5–14, and >14, respectively, among preterm neonates with GAs <28 weeks [3]. These authors subsequently separated LSV into three stages according to the extent and the vessel echogenicity [6], and they found that thin, faintly hyperechogenic LSV (stage 1 LSV) was more likely to be present after PND 15 and that the incidence increased from 13% before PND 4 to 74% after PND 15. They concluded that LSV might be a developmental phenomenon in preterm infant brains. The prevalence of LSV might rise with preterm infant age. Whether preterm neonates have a higher LSV incidence than healthy full-term neonates remains unclear. LSV is not a rare finding during routine cUS examinations of preterm and term neonates NICUs [20]. Makhoul et al. found LSV in 21 of 857 neonates (2.45%) admitted to an NICU, and, of these, 21% were term infants [20]. The findings from a retrospective study of 110 neonates admitted to an NICU, of whom, half were born prematurely, showed that 32 (29.1%) had LSV [21]. However, most of the term infants in these studies had underlying diseases; hence, whether preterm infants are more susceptible to LSV development than term neonates remains unclear, and the actual incidence of LSV in neonates is uncertain.

The definition of and the diagnostic criteria for LSV lack consensus. The most widely accepted definition of LSV is branching or linear hyperechogenic lesions in the basal ganglia and/or thalamus on cUS images but technological advances in cUS may mean that some hyperechogenic lines could be echo reflections from normal lenticulostriate arteries [22,23]; therefore, overinterpretation of echogenic linear lenticulostriate vessels as LSV is possible. Poor interrater agreement is also problematic regarding LSV identification. Sisman et al. undertook a secondary analysis of the prospective results from the Extremely Low Gestational Age Newborns study and showed low levels of agreement among pediatric radiologists regarding LSV diagnoses using cUS; the Kappa values were only 0.18, 0.33, and 0.36 on PNDs 1–4, 5–14, and >15, respectively [3]. We tried to enhance our diagnostic confidence by using two independent readers, and imaging modalities were not performed to confirm LSV diagnoses, because many previous studies’ findings have shown that computed tomography or magnetic resonance imaging failed to reveal corresponding hyperechogenic abnormalities detected using cUS [2,7,24].

LSV tends to appear soon after birth in term infants, but it usually becomes more prominent weeks after birth in preterm neonates [1,2,7,25]. Sisman et al. found that LSV became evident after the second postnatal week in preterm neonates with GAs < 28 weeks [3]. Leijser et al. reported that LSV was first detected in preterm infants at a mean postnatal age of 4–5 weeks [7,25]. Different GAs and cUS screening schedules may alter the postnatal ages at which LSV is first detected. Most investigators agree that the GAs of preterm neonates in whom LSV is detected later are lower than those of neonates with early-onset LSV [7,8,25]. LSV was first detected at a mean postnatal age of 15 weeks in this study, which is much later than was reported previously, but the PMA at which LSV was first detected was term-equivalent age and concurred with that previously reported [7]. We also showed, for the first time, that GA was negatively correlated with the PND on which LSV was first detected.

As with the findings in previous studies, most of our neonates (88.2%) had late-onset LSV [7,10,26]. Sometimes LSV was not detected until 6 months after birth. Postnatal events may have caused late-onset LSV development in most of the neonates. Hemachandra et al. [10] and Leijser et al. [7] found that neonates who developed LSV earlier were more mature than those with late-onset LSV. Leijser et al. also found that the PMA at first detection and the duration of visible LSV did not differ between their early- and late-onset groups [7]. They hypothesized that during the perinatal period, the lenticulostriate vessels are vulnerable to vasculopathy development [7]. We could not confirm this hypothesis, which might have been a consequence of the limited number of neonates with early-onset LSV. We found that the early- and late-onset LSV groups differed significantly regarding the PMA at which LSV was first detected and the visible LSV duration. These results might reflect differences in early- and late-onset LSV development, which may be caused by different predisposing risk factors or pathogeneses.

Previously, LSV was considered an unchanging sonographic finding at birth that did not resolve or progress over time [27,28]. However, our study’s findings showed that LSV progresses and that the mean time to resolution was 6.2 months. Similarly, Weber et al. reported that LSV in four of 11 neonates resolved within 4–7 months [26]. A routine follow-up cUS study finding, from 34 infants, showed that LSV had progressed in 85% of the infants without concurrent clinical deterioration [29]. The authors suspected that the sonographic progression was merely a morphological change without clinical significance that occurred after birth in neonates with prenatal brain insults [29]. Our study’s findings indicated that postnatal insults to preterm neonates’ developing brains may contribute to LSV development and progression.

No consensus exists regarding the antenatal or postnatal risk factors for LSV development in term neonates. While LSV was thought to be initially associated with congenital infections, it is also associated with many noninfectious etiologies, including trisomies, fetal alcohol syndrome, maternal drug exposure, twin-to-twin transfusion syndrome, cardiac disease, neonatal lupus erythematosus, and perinatal asphyxia and hypoxia [26,28,29,30,31,32,33]. A variety of factors can predispose preterm neonates’ brains to LSV. A secondary analysis of data from a randomized controlled trial showed that antenatal exposure to tocolytic MgSO4 ≥50 g was significantly associated with LSV [5]. However, the findings from a prospective study of 21 preterm and term neonates with LSV and 42 matched control infants showed there was no association between antenatal MgSO4 exposure and LSV [20]. Most perinatal characteristics are not associated with neonatal early-onset LSV [7,8], and different postnatal events may contribute to LSV development [3,7,8,27]. We found no associations between the use of MgSO4 and early- and late-onset LSV, but we did not investigate the impact of cumulative antenatal MgSO4 doses.

After adjusting for GA and other confounding factors, we found that longer postnatal oxygen exposure may be associated with a higher risk of late-onset LSV development in preterm infants. This finding has not been previously reported, but hypoxia/ischemia is an important etiologic factor associated with LSV [27]. We could not confirm whether oxygen exposure is an independent factor associated with LSV development or whether prolonged oxygen usage merely reflects the nature of the condition, which is characterized by unstable blood saturation or hypoxia/ischemia. We hypothesized that neonates who require prolonged oxygen therapy may be at a higher risk of hypoxia episodes; therefore, LSV development in neonates exposed to prolonged oxygen therapy may not be caused by the direct effects of oxygen itself, but by the effects of hypoxia and hyperoxia episodes that can induce free radical release and trigger vascular intima injury and LSV development. Further studies are necessary to clarify this.

Congenital CMV infection is the most common infectious cause of neonatal LSV [29,34,35]. In a large retrospective study of 858 newborns, 55 (6.4%) had LSV and, of these, 36.3% were positive for CMV according to urine polymerase chain reactions (PCRs) [36]. Hong et al. found that marked and multilinear LSV is a powerful predictor of CMV infection [36]. Preterm infants are at high risk of CMV infection because they undergo frequent blood transfusions and consume unpasteurized breast milk. Nijman et al. noted that preterm infants who do not have LSV at birth but develop LSV at the term-equivalent age, should be suspected of postnatal CMV infection [37]. The association between CMV infection and LSV was inconclusive in our study, but not all of our enrolled neonates were tested for CMV. Moreover, rather than conducting urine CMV PCRs, we detected CMV using cultures from urine or serum CMV IgM, which may not be as sensitive as CMV PCR.

Our study has several limitations. First, this was a single-center, retrospective observational cohort study. While our cUS examination protocols for each VLBW neonate were uniform, some bias remained. For example, CMV screening was often performed on neonates with clinical symptoms or signs of congenital infection, including SGA, cholestasis, or prominent LSV lesions. CMV infection may have been over-diagnosed in the LSV group; consequently, we could not differentiate between prenatal and postnatal insults-related LSV based on the timing of LSV appearance. Finally, we could not address the impacts of LSV on clinical and neurodevelopmental outcomes.

In conclusion, LSV is not an uncommon finding in cUS scans of VLBW preterm infants. It may be a nonspecific marker of perinatal insults to the developing brains of these infants. The PND on which LSV was first detected was negatively associated with GA. Prolonged oxygen usage may predispose VLBW preterm infants to late-onset LSV development. The long-term clinical impacts of LSV should be investigated further.

## Figures and Tables

**Figure 1 children-08-01166-f001:**
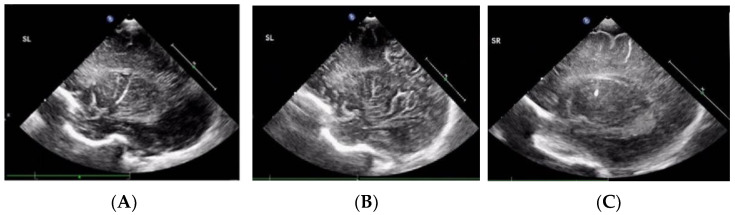
Lenticulostriate vasculopathy over the thalamic/basal ganglia region in three preterm infants. (**A**) Linear type. (**B**) Branching type. (**C**) Punctate type.

**Figure 2 children-08-01166-f002:**
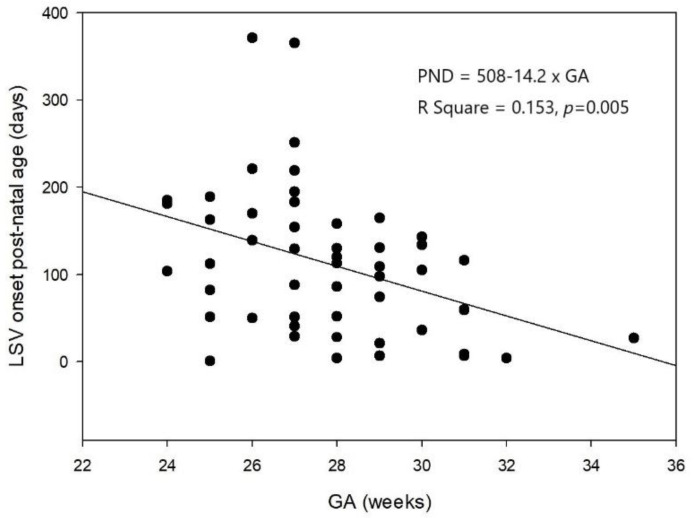
The negative association between the first postnatal days when lenticulostriate vasculopathy was first detected and gestational age analyzed by linear regression analyses (*n* = 51, R^2^ = 0.153, *p* = 0.005).

**Table 1 children-08-01166-t001:** Presentations and evolution of LSV in very-low-birth-weight preterm infants.

Characteristics of LSV	Early-Onset LSV *n* = 6	Late-Onset LSV *n* = 45	Early- vs. Late-Onset LSV *p* Value *
First detection			
Postnatal age (days)	5.5 (1–9)	116 (21–371)	<0.001
PMA (weeks)	31 (25–32)	45 (31–80)	<0.001
Last detection ^#^			
Postnatal age (days)	94 (65–272)	356 (63–494)	0.009
PMA (weeks)	43 (34–69)	78 (37–92)	0.015
Visible duration (days)	72 (62–265)	202 (30–406)	0.042
Appearance at first detection			
Linear	3 (50)	26 (57.8)	1.000
Branching	2 (33.3)	13 (28.9)	1.000
Punctate	1 (16.7)	6 (13.3)	1.000
Laterality			
Unilateral	2 (33.3)	19 (42.2)	1.000
Right	2 (100)	10 (52.6)	0.486
Left	0	9 (47.4)	0.486
Bilateral	4 (66.7)	26 (57.8)	1.000
Associated cUS changes			
IVH (any grade)	2 (33.3)	10 (22.2)	0.616
IVH ≥grade III	0	3 (6.7)	1.000
PVL	1 (16.7)	2 (4.4)	0.319

Continuous variables are presented as the medians (ranges). Categorical variables are presented as numbers (percentages). * The Mann–Whitney U test was used to analyze continuous variables and Fisher’s exact test was used to analyze categorical variables. ^#^ Lenticulostriate vasculopathy was detected in nine neonates at a corrected age of 1 year; these were excluded from the analyses. cUS, cranial ultrasound; IVH, intracranial hemorrhage; LSV, lenticulostriate vasculopathy; PMA, postmenstrual age; PVL, periventricular leukomalacia.

**Table 2 children-08-01166-t002:** Perinatal characteristics of preterm infants with and without lenticulostriate vasculopathy.

	LSV Present *n* = 51	LSV Absent *n* = 79	*p* Value *
Perinatal characteristics			
Boy:girl, *n*	30:21	42:37	0.526
GA (weeks) ^‡^	27.8 (2.3)	28.4 (2.3)	0.150
BBW (g)	1070 (253)	1114 (243)	0.329
Vaginal delivery	4 (7.8)	10 (12.7)	0.387
Twin gestation	15 (29.4)	26 (32.9)	0.657
SGA	6 (11.8)	19 (24.1)	0.083
LGA	2 (3.9)	3 (3.8)	1.000
Antenatal steroid use	37 (72.5)	58 (73.4)	0.913
Antenatal MgSO_4_ use	10 (19.6)	23 (29.1)	0.224
Maternal age	33.4 (5.0)	34.2 (4.8)	0.390
History of pre-eclampsia	10 (19.6)	18 (22.8)	0.667
History of PPROM	18 (35.3)	21 (26.6)	0.290
History of GDM	7 (13.7)	14 (17.7)	0.546
Apgar score <7 at 5 min	15 (29.4)	19 (24.1)	0.497
Placental findings ^‡^			
HCAM	24 (57.1)	35 (50.7)	0.560
Intermediate-to-advanced HCAM	18 (42.9)	17 (24.6)	0.045
Funisitis	4 (9.5)	8 (11.6)	1.000
Laboratory results			
Maternal WBC count (×10^3^/µL)	14.82 (0.53)	13.12 (0.46)	0.060
Maternal CRP (mg/dL)	1.35 (1.7)	1.48 (1.9)	0.741
Neonatal WBC count (×10^3^/uL) ^||^	9.71 (0.45)	10.05 (0.78)	0.777
Neonatal CRP (mg/dL) ^||^	0.53 (0.95)	0.54 (0.89)	0.961
Clinical outcomes			
RDS	27 (52.9)	34 (43)	0.269
PPHN	4 (7.8)	8 (10.1)	0.763
Pneumothorax	5 (9.8)	6 (7.6)	0.751
BPD	17 (33.3)	15 (18.9)	0.064
Steroid use for BPD	6 (11.8)	2 (2.5)	0.056
Early onset sepsis	0	1 (1.3)	1.000
Late onset sepsis	17 (33.3)	15 (18.9)	0.064
Mechanical ventilator usage duration (days)	8.2 (13.8)	4.4 (6.8)	0.073
Oxygen usage duration (days)	49 (1–129)	38 (4–129)	0.018
PDA	16 (31.3)	26 (32.9)	0.855
ROP	2 (3.9)	6 (7.6)	0.480
NEC	2 (3.9)	8 (10.1)	0.314
IVH ≥ grade III	3 (5.9)	5 (6.3)	1.000
PVL	3 (5.9)	7 (8.9)	0.739
CMV infection ^#^	4 (18.2)	3 (21.4)	0.567

Continuous variables are presented as the mean and the standard deviations. The categorical variables are presented as numbers (percentages). * Independent Student’s *t*-tests and χ^2^ tests were performed on continuous and categorical variables. ^‡^ There were 42 and 69 placenta pathology reports available for the lenticulostriate vasculopathy present and lenticulostriate vasculopathy absent groups, respectively. ^||^ Neonatal white blood cell count at birth and serum C-reactive protein level at 24 h after birth. ^#^ There were 22 and 14 cytomegalovirus urine isolations and cytomegalovirus immunoglobulin M reports available for the lenticulostriate vasculopathy present and lenticulostriate vasculopathy absent groups, respectively.

**Table 3 children-08-01166-t003:** Multivariate analysis of the risk factors associated with lenticulostriate vasculopathy in very-low-birth-weight preterm neonates (*n* = 130).

	Odds Ratio	95% CI	*p* Value
GA (increase 1 week)	1.099	0.777, 1.555	0.592
Intermediate to advanced HCAM (yes vs. no)	1.857	0.784, 4.398	0.159
Oxygen usage duration (increase 1 day)	1.023	0.996, 1.050	0.098

CI, confidence interval; GA, gestational age; HCAM, histological chorioamnionitis.

**Table 4 children-08-01166-t004:** Perinatal characteristics of the early-onset lenticulostriate vasculopathy (LSV), late-onset LSV, and non-LSV groups, and comparisons between the groups regarding clinical parameters.

Clinical Parameters	LSV Absent *n* = 79	Early Onset LSV *n* = 6	Late Onset LSV *n* = 45	Early ^†^ vs. Late Onset LSV *p* Value	Early ^†^ Onset vs. LSV (-) *p* Value	Late * Onset vs. LSV (-) *p* Value
Male	42 (53.2)	2 (33.3)	28 (62.2)	0.214	0.432	0.328
GA (weeks)	28.49 (2.3)	30 (25–32)	27.6 (2.2)	0.073	0.283	0.061
BBW (g)	1114 (243)	1253 (836–1455)	1051 (254)	0.165	0.272	0.173
Vaginal delivery	10 (12.7)	1 (16.7)	3 (7.7)	0.404	0.576	0.265 *
Twin gestation	21 (32.9)	0	15 (33.3)	0.162	0.171	0.962
SGA	19 (24.1)	2 (33.3)	4 (8.9)	0.141	0.644	0.037
LGA	3 (3.8)	0	2 (4.4)	1.000	1	1 *
Antenatal steroid usage	58 (73.4)	4 (66.7)	33 (73.3)	0.661	0.660	0.992
Antenatal MgSO_4_ usage	23 (29.1)	1 (16.7)	9 (20)	1.000	0.671	0.265
Pre-eclampsia	18 (19.6)	3 (5)	7 (15.6)	0.081	0.157	0.335
GDM	14 (13.7)	1 (16.7)	6 (13.3)	1.000	1	0.523
PPROM	21 (35.3)	1 (16.7)	17 (37.8)	0.405	1	0.194
HCAM ^‡^	35 (50.7)	3 (60)	21 (56.8)	1	1	0.553
Funisitis ^‡^	8 (11.6)	1 (20)	3 (8.1)	0.410	0.487	0.744
Apgar score <7 at 5 min	19 (24.1)	1 (16.7)	14 (31.1)	0.657	1	0.392
RDS	34 (43)	2 (33.3)	25 (55.6)	0.402	1	0.180
Sepsis	16 (20.3)	2 (33.3)	15 (33.3)	1.000	0.604	0.115
PDA	26 (32.9)	2 (33.3)	14 (31.1)	1.000	1	0.837
BPD	15 (18.9)	3 (50)	14 (31.1)	0.387	0.106	0.125
ROP	6 (7.6)	1 (16.7)	1 (2.2)	0.224	0.413	0.400
NEC	8 (10.2)	0	2 (4.4)	1.000	1	0.264
CMV ^§^	3/14 (21.4)	0	4/19 (21)	1.000	1	1
Mechanical ventilator usage duration (days)	4.4 (6.8)	0 (0–21)	8.7 (14.3)	0.138	0.359	0.061
Oxygen usage duration (days)	38 (4–129)	32.5 (8–101)	49 (1–129)	0.430	0.945	0.010

The continuous variables are presented as the mean and the standard deviations. The categorical variables are presented as the numbers (percentages). * Independent Student *t*-tests and χ^2^–tests were performed for continuous and categorical variables. ^†^ Mann–Whitney U test and Fisher’s exact test were used for continuous and categorical variables. ^‡^ There were five and 39 placenta pathology reports that were available for the early-onset lenticulostriate vasculopathy and late-onset lenticulostriate vasculopathy groups, respectively. ^§^ There were three and 19 CMV urine isolation results or CMV IgM reports were available in the early-onset LSV and late-onset LSV groups, respectively.

## Data Availability

Data is contained within the article.

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
