# Peer review of "Lenticulostriate Vasculopathy in Very-Low-Birth-Weight Preterm Infants: A Longitudinal Cohort Study"

_children, 2021, doi:10.3390/children8121166_

Round 1
Reviewer 1 Report
Hung et al. aimed to describe the prevalence, presentation, and evolution of LSV, and the perinatal risk factors of LSV among VLBW preterm infants up to 1 year of corrected age, in a retrospective single center study. To my opinion the study is well described, well designed and innovative. It has some limitation (i.e. the retrospective design and the absence of long-term neurodevelopment data) that are well descripted in the discussion section. I found some confusion in the statistical section and results presentation, that should be resolved during the revision process before the publication.
1) First of all, I suggest an important English revision to improve the rideability of the study.
2) In methods section I suggest to add a reference for the definition of all prematurity related conditions. In addition, in the text you talk about BPD, in the tables CLD, please use the same denomination.
3) line 69- the name of the NICU where the newborns were admitted should be added.
4) lines 139-143. It is not completely clear for me. You performed a correlation between days of LSV onset and GA at birth, PMA of LSV onset and GA at birth. Ad after you performed a multivariate analysis using as dependent variables LSV disappearance PNDs or at particular PMAs and as covariates GA, weight, or sex? Or is it maybe a correlation with the single variables too? In this case, why did you correlate days of LSV disappearance also with weight or sex and not the same for LSV diagnosis? Statistical methods and results presentation are not clear, please revise.
5) I don’t understand why you performed two differently multivariate analysis. I suggest to perform a unique binary regression analysis with all the statistically significant variables in the univariate analysis. In addition, keep attention. GA and duration of oxygen therapy is not a categorial variables. In logistic regression analysis covariates and dependent variables should be categorial. I suggest to found a cut off for each variable (by example GA < 28 weeks of PMA and the same for “prolonged oxygen administration”).
6) Please modified the tables, especially table 1. I read with extreme difficulty the table, that is not clear. Also, table legend should be presented in accordance with Journal guidelines for each table (https://www.mdpi.com/journal/children/instructions).
7) Please modify the text in accordance with Journal guidelines. No running title are needed, abbreviations should be defined the first time they appear in each sections of the text (no in a separate paragraph) and mails and correspondence are not written as suggested (https://www.mdpi.com/journal/children/instructions).
Reviewer 2 Report
In this paper, Hung et al reported LSV may be a nonspecific marker of perinatal insults to preterm infants’ developing brains. Prolonged postnatal oxygen usage may predispose VLBW preterm infants to late-onset LSV development.
The relationship between oxygen administration and various other parameters and LSV is being investigated.
Multivariate analysis has also been conducted to identify important factors and risk factors.
Conducting the analysis for as long as seven years is a difficult task, and we believe it contains important findings that will be useful in the future.
The limitation is that it is a single-center, retrospective observational cohort study. They could not differentiate between prenatal and postnatal insult-related LSV based on the timing of LSV appearance. They could not address the impacts of LSV on clinical and neurodevelopmental outcomes.
I would like you to report on the effects on neurodevelopment in future reports.
Reviewer 3 Report
Hung and colleagues present a well considered and undertaken retrospective study to determine the prevalence, presentation, and evolution of LSV, and perinatal risk factors associated with LSV among very-low-birth-weight preterm infants.
It is an important study with findings that will provide a reference point for future investigations on LSV. Manuscript is well written with minor editing required e.g.
Ln 54-55
Antenatal high dose of MgSO4 has been showed to be associated with LSV in preterm neonates in a large randomized control trial cohort study [5]. Please rephrase.
I believe this study to be of importance to the paediatric community and as such will pique the interest of clinical specialists in this area.
Round 2
Reviewer 1 Report
Thank you for your precise revision.
Congratulation!